# A method for using stationary networks to observe long term trends of on-road emissions factors of primary aerosol from heavy duty vehicles

Helen L. Fitzmaurice[1], Ronald C. Cohen[1,2]

[1]Department of Earth and Planetary Science, University of California Berkeley, Berkeley, 94720, United States
[1]Department of Chemistry, University of California Berkeley, Berkeley, 94720, United States

*Correspondence to*: Ronald C. Cohen (rccohen@berkeley.edu)

**Abstract** Heavy-duty vehicles (HDV) contribute a significant, but decreasing, fraction of primary aerosol emissions in urban areas. Previous studies have shown spatial heterogeneity in compliance with regulation. Consequently, location-specific emissions factors are necessary to describe primary particulate matter (PM) emissions by HDV. Using near-road observations from the Bay Area Air Quality Management District (BAAQMD) network over the 2009-2020 period in combination with Caltrans measurements of vehicle number and type, we determine primary $PM_{2.5}$ emission factors from HDV on highways in the San Francisco Bay Area. We demonstrate that HDV primary aerosol emission factors derived using this method are in line with observations by other studies, that they decreased a by a factor of ~9 in the past decade, and that emissions at some sites remain higher than would be expected if all HDV were in compliance with California HDV regulations.

## 1 Introduction

Exposure to aerosols smaller than 2.5 microns in diameter ($PM_{2.5}$) at current ambient levels is estimated to cause 130,000 excess deaths per year in the United States (Tessum et al., 2019). Epidemiological studies have shown that health and mortality impacts from $PM_{2.5}$ persist at concentrations of $PM_{2.5}$ below current National Ambient Air Quality Standards and that small changes in $PM_{2.5}$ concentration may result in substantial health impacts (Di et al., 2017). Because of the health impacts resulting from small increases in $PM_{2.5}$, air quality academics, public health researchers, local regulatory agencies, and state governments have come to appreciate the importance of neighborhood scale differences in cumulative exposure to $PM_{2.5}$ (e.g. CARB, 2018). For example, regulatory agencies in California have begun to shift from a paradigm based primarily on compliance with annual and daily, regional scale air quality metrics to one also focused on mitigation of cumulative exposure, creating local remediation plans based on source apportionment. (BAAQMD, 2019). These source apportionment estimates are created from bottom-up emissions inventories using emissions factors and activity data. Consequently, accurate local emissions factors are vital to understanding and planning neighborhood-scale mitigation strategies.

On-road vehicles, specifically HDV, are a large contributor to aerosol in urban areas, both through direct emissions and through secondary formation in the atmosphere (e.g. Shah et al., 2018; BAAQMD, 2011). Total emissions can be thought of

as the product of emissions factors (EFs) and the activity, where the EFs are expressed in units of grams of aerosol per unit activity (such as grams of aerosol per kg of fuel burned or per km travelled). EFs are estimated for on-road activity in a

variety of ways including scaling based on measurements in a lab setting and/or on-road measurements (See references, Table 1). A summary of on-road studies for primary HDV and passenger vehicle $PM_{2.5}$ EFs over the last 25 years is shown in Figure1 and Table 1. These studies determined EFs of primary on-road aerosol by comparing ratios of aerosol enhancement (in grams) to $CO_2$ and/or CO enhancement (as a measure of fuel burned). Measurements included sampling directly in the exhaust of tunnels, and high frequency sensors near or above roads to sample and characterize individual vehicle plumes.

These prior observations show that typical heavy duty, diesel-powered vehicles dominated on-road emissions of primary aerosol in the 1990s and early 2000s. However, in recent years, emissions factors from typical heavy-duty vehicles have been dramatically reduced such that $PM_{2.5}$ EFs of HDVs are now similar to those of light duty vehicles (LDV) and are less than 0.05 g $PM_{2.5}$ /kg fuel burned. Control technologies such as diesel particulate filters and selective catalytic reduction are contributing to these reductions in EFs for HDVs.

While these improvements are seen in the "typical" HDV, previous studies indicate that compliance of HDV with emission technology requirements, and therefore HDV on-road emissions factors, vary by up to an order of magnitude from location to location (Preble et al., 2018; Bishop, 2015; Haugen et al., 2018; Haugen et al., 2019). For example, Bishop (2015) and Haugen et al., (2018, 2019) found emissions factors measured at the Port of Los Angeles were as much as an order of magnitude lower than those measured along a highway in Cottonwood, California during the same season. While the gap

between the two sites narrowed from 2013-2017, the mean emission factors measured in Cottonwood were still 3 times those measured at the Port of Los Angeles in 2017. Similarly, Preble (2018) found that while 100% of trucks at the Port of Oakland were registered by the state of California as being in compliance with HDV control technology regulations, compliance rates amongst HDV at the Caldecott tunnel (also in Oakland, CA) were below 90%.

These studies highlight that variability in emissions factors as a function of location may affect exposure. They point to the

importance of characterizing spatial variation in HDV emissions if we are to understand aggregate emissions from the sector and its localized impacts. To assess the potential for existing data sources to supply the needed information, here we explore the use of regulatory sensor networks (near-highway, hourly $PM_{2.5}$ and CO (or $CO_2$) measurements), paired with coincident traffic data including LDV and HDV counts, to quantify spatial variation in HDV EFs. Such data is widely available. For example, in the US, there are more than 550 regulatory sites at which $PM_{2.5}$ and CO are collocated, some of which have

measurements spanning more than a decade (https://www.epa.gov/outdoor-air-quality-data). Of these, 154 are located within 500 m of a highway. The large number of these sites and their longevity allow for examination of regional and temporal differences in EFs for HDV across the United States. In the future, the approach we outline should be even more widely applicable when dense low-cost sensor networks including aerosol and CO or $CO_2$ are available as a data source (e.g. Shusterman et al., 2016, Kim et al., 2018; Zimmerman et al, 2018). Because HDV emissions control regulations vary

regionally in the US, this method has the potential to shed light on regional differences in HDV EF trends.

We begin by describing a general method for using such data to derive EFs of primary PM$_{2.5}$ from HDV (Section 2). We then (Section 3) test our method by using data from four near-highway sites operated by the Bay Area Air Quality Management District (BAAQMD) in the San Francisco Bay Area (Figure 2a) over the period of 2009-2018. In section 4 we discuss the relationship of these findings to measures of exposure.

**2 Data and Methods**

2.1 Aerosol and CO Measurements

We use 1 hr averaged observations from 18 of the BAAQMD regulatory sites which measure PM$_{2.5}$ using Beta Attenuation Monitors and CO using the Thermo Scientific TE48i IR sensor. Some sites have been in operation since 2009, while others have been brought online as recently as 2018, or were operational for only a few years during this time period. Data was

retrieved from https://aqs.epa.gov/aqsweb/documents/data_api.html. Site locations are summarized and example data are shown in Figure 2. PM$_{2.5}$ and CO data from four near-highway sites (San Rafael, Redwood City, Berkeley Marina, Pleasanton) are used to characterize $EF_{PM(HDV)}$, and data from other sites are used to define regional signals.

2.2 Meteorology

Boundary layer height and wind speed and direction were retrieved from the European Center for Meteorology and Weather Forecasting (ECMWF) ERA5 reanalysis, (https://cds.climate.copernicus.eu/cdsapp#!/dataset/reanalysis-era5-land?tab=form). Typical diel cycles for boundary layer height and total windspeed are shown in Fig. S1.
We use this reanalysis to find windspeed, boundary layer height, and wind direction for each hour (2009-2020) at each of the BAAQMD sites. Wind data is then used for filtering PM$_{2.5}$ and CO measurements as described below.


2.3 Traffic Data

Total vehicle flow, fleet speed, and the percent of vehicles that are HDV are taken from the Caltrans' Performance Measurement System (PeMS) database (http://pems.dot.ca.gov), which records these parameters at over 1800 locations on highways in the Bay Area. We include all BAAQMD sites that are within 500 meters of one major highway and use traffic

count data from the PeMS measurement site closest to each air quality site. In cases of missing PeMS data, data was filled in with the median value associated with that parameter for a particular site in a particular year, or if not possible, retrieved from the second or third nearest sites. More details about the PeMS data including a map of PeMS measurement sites, a list of sites used in this study, and example diels of truck flow and truck percent are presented in Fig. S2, Tbl. S1, and Fig. S3.

2.4  The EMissions FACtor (EMFAC2017) Model

In order to calculate EF$_{PM(HDV)}$ as describe in section 2.5, we make use of the EMFAC2017 model to estimate both HDV and LDV emission factors for CO, as well as a HDV and LDV fuel efficiency. We run this model for the four time periods of

interest (2009-2011, 2012-2014, 2015-2017, 2018-2020) by choosing the middle year for that period, specifying location to be the nine counties under BAAQMD's jurisdiction. We assign vehicle class to either LDV or HDV by approximate vehicle length, as this is the manner in which PeMS classifies vehicles as either LDV or HDV. These designations are summarized in the supplement of Fitzmaurice, et al. (2022). We use EMFAC emission values across all speeds to obtain CO emission factor used to calculate $EF_{PM(HDV)}$ for all sites during a given time period.

To estimate uncertainty in hese emission factors at particular sites, we use speed-dependent variance in EMFAC-derived emission factors. To do this, we first calculate speed-dependent CO emission factors (g CO / kg fuel), as well as emission rates g $CO_2$ / vkm for HDV and LDV for each time period as follows:

$$X_{speed,HDV/LDV} = \frac{\sum_{i=1}^{n} vkm_{i,speed}X_{i,speed}}{\sum_{i=1}^{n} vkm_{i,speed}}. \quad (1)$$

Here, $vkm$ is the EMFAC model's estimate of kilometers traveled per year by a particular vehicle class, $X$ is either emission rate (g $CO_2$ / vkm) or emission factor in (g CO / kg fuel). The EMFAC2017 model bins speeds (5 mph each), so we use spline interpolation to estimate CO emission factor and emission rate hourly at each PeMS site corresponding to a BAAQMD site of interest. The $1\sigma$ variance of these estimates during times corresponding to those used to calculate $EF_{PM(HDV)}$ are then used estimate uncertainty in emission rate and CO emission factors. These in turn are used to estimate uncertainty in $EF_{PM(HDV)}$.

## 2.5 Derivation of $EF_{PM(HDV)}$

Our derivation of HDV EFs assumes that the relationship between the enhancement of $PM_{2.5}$ and CO, as observed near-road, can be scaled so that it represents PM per unit of fuel burned by HDVs:

$$EF_{PM(HDV)} = \gamma \frac{PM_{HDV}}{CO_{fleet}} \frac{g\,CO_{,fleet}}{kg\,fuel_{,HDV}}, \quad (2)$$

In this equation, $\gamma=0.0008$, and is the ideal gas law conversion factor, from ($\mu g/m^3 ppm^{-1}$-CO) to ($g\,PM_{2.5}/gCO$). A detailed derivation of Eq 2. is described in Sect. S3. Below, we describe the steps used to calculate each term in equation (2).

The first term $\frac{PM_{HDV}}{CO_{fleet}}$ in the equation is derived from observations as the slope of a linear fit of near-road $PM_{2.5}$ (assumed to be primarily emitted by HDV) and near-road CO (assumed to be emitted by both HDV and LDV). This term is derived by (1) isolating local enhancements of $PM_{2.5}$ and CO, (2) isolating roadway enhancements by use of temporal and meteorological filters and (3) fitting resulting roadway enhancements of $PM_{2.5}$ and CO to a line, as detailed below.

(1) To isolate local enhancements from total signal $PM_{2.5}$ and CO, we first leverage the entire BAAQMD network to derive an hourly regional signal for each species. The regional signal is defined as the $10^{th}$ percentile of the data across all 22 BAAQMD sites within a five-hour window of that hour (Figure 2b). We choose the bottom $10^{th}$ percentile rather than the absolute minimum in hopes that the baseline captures regional mixing rather than just

cleaner "background" air. In Sect. S4, we show that while the $EF_{PM(HDV)}$ is slightly sensitive to the percentile and time-window chosen, sensitivity of $EF_{PM(HDV)}$ to these parameters is smaller than estimated uncertainty in final $EF_{PM(HDV)}$ values. This regional signal is assumed to be composed of background $PM_{2.5}$ /CO transported to the region from elsewhere as well as region-wide sources of secondary aerosol/CO. We the find the enhancement by local primary emissions by subtracting the regional signal from total signal at each site.

(2) We isolate primary emissions from on-road sources by considering only the morning commute times and only during fall and winter and applying meteorological filters. These are times coinciding with relatively high traffic emissions and too early in the day for significant accumulation of new secondary aerosol. We find the 6-8 am period represents the optimal overlap of low boundary layer height (Figure S1) and HDV emissions (Figure S3). The combination of low boundary layer height and stable early morning conditions enhance signal (Choi et al., 2012; Choi et al., 2014), allowing inferences about traffic from sites further away than would be possible during later morning or afternoon.

To avoid observations of stagnant air, we only include observations with wind speed above 0.5 m/s. Furthermore, for each site of interest, exclude known fire events and we filter out observations that occur when the BAAQMD site is upwind of the highway. An upwind event is defined as when the wind direction deviates more than 90 degrees in either direction from the perpendicular line pointing from the highway nearest a BAAQMD site to that site. The result of these first two steps are enhancements in $\Delta PM_{2.5}$ and $\Delta CO$ above background.

(3) The slope of a linear fit of all unfiltered $\Delta PM_{2.5}$ and $\Delta CO$ (see figure 3) is defined as the "enhancement ratio," in units of $\mu g/m^3 ppm^{-1}$-CO. Using the lengthy dataset, we can derive enhancement ratio for different percentages of HDV in the vehicle fleet on the road. There are some high $\Delta PM_{2.5}$ values uncorrelated with $\Delta CO$ as shown in Figure 3. In all cases, these points show little to no $NO_x$ enhancement and thus are characteristic of a source that is not HDV. We make the assumption that LDV $PM_{2.5}$ EFs are negligible and on-road primary emissions of aerosol are solely from HDV, implying that the enhancement ratio is equivalent to the term $\frac{PM_{HDV}}{CO_{fleet}}$. We discuss this assumption and the impact of correcting for LDV emissions further in Sect. 3.

The term $\frac{g\ CO_{fleet}}{kg\ fuel_{HDV}}$ is can be calculate using HDV fraction, $t$, and LDV and HDV CO emission factors (g CO / kg fuel) and emission rates (g $CO_2$ / km) from EMFAC2017 model as follows:

$$\frac{g\ CO_{fleet}}{kg\ fuel_{HDV}} = \frac{EF_{CO(HDV)}tE_{HDV} + EF_{CO(LDV)}(1-t)E_{LDV}}{tE_{HDV}}, (3)$$

where $t$ is the HDV fraction and $E$ is emission rate.

Because, at a given site, we expect $\frac{PM_{HDV}}{CO_{fleet}}$ (but not $EF_{PM(HDV)}$) to vary linearly with HDV fraction, we bin data by HDV fraction in increments of 0.02, and use the process above to calculate $EF_{PM(HDV)}$ for each bin. Data and slopes for each bin are shown in Fig. S6. We then calculate $EF_{PM(HDV)}$ for each site during a particular time period using the average of the,

weighted by uncertainty in $EF_{PM(HDV)}$ for each bin. A detailed description of how we estimate the uncertainty in $EF_{PM(HDV)}$ for each bin can be found in Sect. S6.

## 3 HDV Emissions Factors from Primary Aerosols in SF Bay Area: 2009-2020

The result of this procedure is $EF_{PM(HDV)}$ at four near-highway BAAQMD sties (Redwood City, Berkeley Marina, San Rafael, Pleasanton) during the time periods: 2009-2011, 2012-2014, 2015-2017, 2018-2020 (Fig. 4). We observe $EF_{PM(HDV)}$ decrease substantially over the decade (Fig. 1, Fig. 4), amounting to a roughly nine-fold reduction. We also observe substantial site to site differences in $EF_{PM(HDV)}$. For example, during the 2018-2020 period, we see a range of a factor of ~7 of 0.05+/-0.06 g $PM_{2.5}$ / kg fuel to a maximum of 0.35 +/- 0.08 g $PM_{2.5}$ / kg fuel. In addition, we observe different timing

emission factor decreases between sites (e.g. Redwood City and San Rafael). For example, while emission factors at both Redwood City and Santa Rosa drop throughout the time period, values at San Rafael in the 2018-2020 time period are similar to those seen at Redwood City in 2012-2014, suggesting a difference in timing of compliance to control technologies at each place. Both the temporal decrease and the site-to-site differences in $EF_{PM(HDV)}$ are similar to prior reports derived using other approaches to data collection and interpretation (e.g. Haugen et al. 2017, 2018).

In addition to being in line with observations from other studies, the observed decrease in $EF_{PM(HDV)}$ follows progressively more stringent truck regulations by the state of California over that time. However, in the 2018-2020 period, observed $EF_{PM(HDV)}$ are still higher than would be expected were all vehicles in compliance with California regulations. By 2020, California law required that all HDV models from the years 1995-2003 replace their engines with 2010 or newer models, and that all HDV model year 1994 or newer use diesel particulate filters (DPF) (California Code of Regulations). Assuming that

the fleetwide average EF for models with 2010 or newer engines using DPF is 0.03 g PM / kg fuel as observed by Haugen (2018), we can use fuel usage by HDV model year in 2020 as well as emissions factors for vehicles older than 1994 estimated by the Emissions FACtor Model (EMFAC2017) to estimate a fleetwide average. Thus a fleetwide average should have an EF of 0.03-.06 g PM / kg fuel if the trucks were fully compliant in 2018-2020. In contrast, we observe an average EF of 0.08 +/-0.03 g $PM_{2.5}$ / kg fuel, for 2018-2020.  While our estimates overlap with the higher end of what is expected

counting uncertainty, it is larger than expected for an HDV fleet compliant with current regulations. Non-exhaust vehicle emissions (e.g. tire wear, brake wear) may account for some of this discrepancy. However, we observe substantially higher emission factors at highways near the Pleasanton (0.35 +/- 0.08 g PM / kg fuel) and Berkeley Marina (0.15 +/ 0.12 g $PM_{2.5}$ / kg fuel) sites. Possible explanations for this discrepancy include exemptions from truck regulations, under which certain classes of HDV travelling less than 15,000 miles per year are eligible for exemptions, meaning locally

travelling HDV may have higher emissions factors than those travelling long distances (CARB, 2018), the fact that HDV registered in other states are not typically subject to CA regulations unless they enter specific areas, such as ports, and failure of or tampering with installed equipment.

In considering estimated $EF_{PM(HDV)}$, it is important to consider two potential biases in our method: the impact of $PM_{2.5}$ emitted by LDV and the potential for local sources to bias emissions estimates. As stated in Sect. 2, in calculating $EF_{PM(HDV)}$,

we assume that contribution of $PM_{2.5}$ from LDV is negligible. This assumption is sound at the beginning of our period (2010s) of interest, because reported values of $EF_{PM(HDV)}$ were more than an order of magnitude higher than $EF_{PM(LDV)}$ at that time (Fig 1). More recently, as $EF_{PM(HDV)}$ has decreased this is less clear, especially without on-road estimates of $EF_{PM(LDV)}$, and because LDV also contribute non-tailpipe emissions of $PM_{2.5}$ from brake and tire wear. However, for 2020, EMFAC still estimates the ratio of $EF_{PM(HDV)}$ : $EF_{PM(LDV)}$ to be ~8. Such a ratio would mean that even if only 5% of vehicles were HDV,

more than 60% of $PM_{2.5}$ emissions are expected to be attributable to HDV. This is an important concern, and we address it in two ways. First, we show that even in the 2018-2020 period, the PM:CO enhancement ratio increases with HDV % regardless of total flow rate (Fig. 5, left). We interpret the intercept of a linear fit with these data to be the $PM_{2.5}$ resulting from LDV alone and note it to be much smaller than the impact of increasing HDV by only a few percent. The observed $PM_{2.5}$:CO intercept would correspond to an $EF_{PM(LDV)}$ to be ~0.01g $PM_{2.5}$ / kg fuel. This value is roughly consistent with tire

and brake emission factors from EPA MOVES3 (EPA, 2020), although it is difficult to know the extent of braking at a given site, and estimates form previous studies of non-exhaust $PM_{2.5}$ by LDV vary widely (Fussell et al., 2022). Second, we explore the impact that LDV emissions might have on $EF_{PM(LDV)}$. To understand the impact of LDV $PM_{2.5}$ emissions on our findings, we assume $EF_{PM(LDV)}$ to be 0.01g PM / kg fuel and recalculate $EF_{PM(HDV)}$. As shown in Fig. 5, right, correction for LDV emissions in this way decreases estimated $EF_{PM(LDV)}$, bringing the average value in the 2018-2020 period to 0.03, which

is in line with what would be expected if all Bay Area HDV were in compliance with regulations during that period. However, even after this correction, $EF_{PM(HDV)}$ at Pleasanton and Berkeley Marina are still substantially higher (0.32+/-0.08 and 0.13 +/- 0.05) than would be expected if all HDV were compliant.

The second potential for bias in the method presented here is the influence of local, non-highway sources on measured $PM_{2.5}$ and CO enhancements. Because our method is dependent on finding the slope of $PM_{2.5}$ and CO, we expect this to eliminate

contributions from non-combustion sources for which $PM_{2.5}$ and CO are uncorrelated. However, nearby combustion, such as non-highway vehicle sources, has the potential to influence $EF_{PM(HDV)}$ results. For example, we consider the $EF_{HDV}$ calculated for Laney College, a near-highway BAAQMD site not considered in the analysis above. The Laney College site instruments are located in a large parking lot. In the 2015-2017 and 2018-2020 periods is significantly higher than $EF_{HDV}$ observed at the four sites we deem reliably far from other sources. While it is possible that HDV on the highway near Laney

College are unusually high emitters, it is more likely that emissions from a nearby parking lot are responsible for the high inferred EFs. This is because $PM_{2.5}$:CO emissions ratios are expected to be dramatically higher at low (parking lot) speeds compared to speeds typically seen on a highway, meaning that a comparatively small number of vehicles may contribute significantly to $PM_{2.5}$:CO enhancement ratios. (See Sect. S8, Fig. S8.) This example shows that while the method developed in this paper has the potential to leverage existing data to highlight potential hotspots for $EF_{PM(HDV)}$ non-compliance, care

must be taken in interpretation of resulting emission factors.

**4 - Primary $PM_{2.5}$ exposure**

To understand exposure from HDV PM$_{2.5}$, we calculate both a region-wide addition to aerosol burden by HDV emissions and an enhancement as a function of distance from a highway. Assuming steady-state, a box of 100 km in length, 160 m in height, and a wind-speed of 1.2 m/s (Figure S6), and using fuel sales data (Moua, 2020) to estimate total HDV fuel used, we estimate a maximum region-wide enhancement on the order of 0.12 $\mu$g/m$^3$ on a typical day in the 2018-2020 period, compared to an enhancement of 1.1 $\mu$g/m$^3$ during the 2009-2011 period (Figure S7). Decreases in emissions factors over the past decade are countered by the increase in diesel fuel usage (70%) (Moua, 2020) such that there has been only a small change in typical regional exposure to primary PM from HDV. (See Fig. S1 for diel cycle of modeled region-wide enhancement.) While an enhancement of 0.12 $\mu$g/m$^3$ is small in comparison to average ambient PM$_{2.5}$ (8.3-14.4 $\mu$g/m$^3$ for all BAAQMD sites in 2018), it is sizeable in comparison to average ambient BC (.4-1 $\mu$g/m$^3$ for all BAAQMD sites in 2018). To gauge near roadway exposure, PM$_{2.5}$ enhancement from HDV was calculated as a function of distance from a highway, modeled treating emissions from the highway as a gaussian plume flowing perpendicular to a line source. Assuming both highway and point of measurement at ground level, the simplified gaussian plume dispersion for a line source yields:

$$PM_{2.5\,enh} = \frac{2\lambda}{\sqrt{2\pi}u\sigma_z} \quad (3)$$

where $\lambda$ is an emissions rate per unit highway length, u is wind speed, and $\sigma_z$ is a dispersion parameter. Using the average emission factor from the 2018-2020 time period, for a typical daytime HDV flow rate of 500 vehicles per hour (Figure S2) and windspeed of 1.2 m/s (Figure S6), we calculate PM$_{2.5}$ enhancement as a function of perpendicular distance downwind of a highway. For unstable atmospheric conditions ($\sigma_z = \frac{0.102x}{\left(1+\frac{x}{927}\right)^{-1.92}}$), enhancements drop to values of below 0.8 $\mu$g/m$^3$ in the first 200 m. For stable conditions ($\sigma_z = \frac{0.022x}{\left(1+\frac{x}{1170}\right)^{0.7}}$), such as those typical of early morning, enhancements of 1 $\mu$g/m$^3$ are predicted up to a kilometer away.

**5 Conclusions**

We find that HDV EFs in the SF Bay Area have decreased by about a factor of ~9 over the last decade, consistent with trends reported in other analyses in this region and Los Angeles. We find spatial variation of HDV EFs remains large indicating a wide range in the application of retrofit technologies and the possibility that vehicles legally exempt from compliance with the current standards are a significant portion of those on the road at the sampling sites. The method introduced in this paper has the potential to leverage existing regulatory (or other) data to examine long-term trends and highlight potential spatial heterogeneities in $EF_{PM(HDV)}$.

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

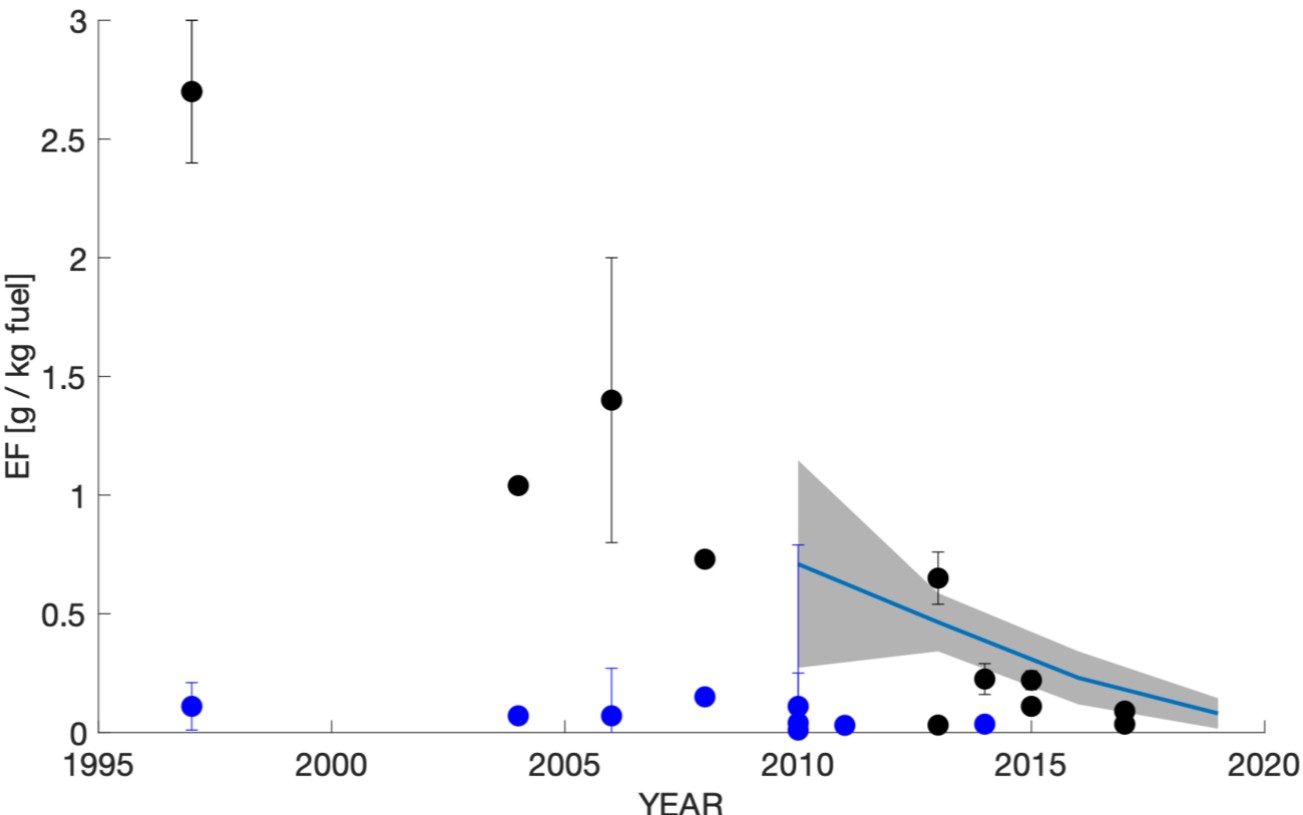

**Figure 1:** On-road measurements of emissions factors, from other studies. HDV (black) emissions factors converge on LDV (blue) emissions factors. Some studies do not give error bars. Grey patches and blue trendline indicate findings from this study for the two highway sites (RWC and SR) available during all time periods. Blue trendline shows the error-weighted mean of emission factors at these two sites during each time period. Grey patches indicate the estimated uncertainty in the error-weighted mean.


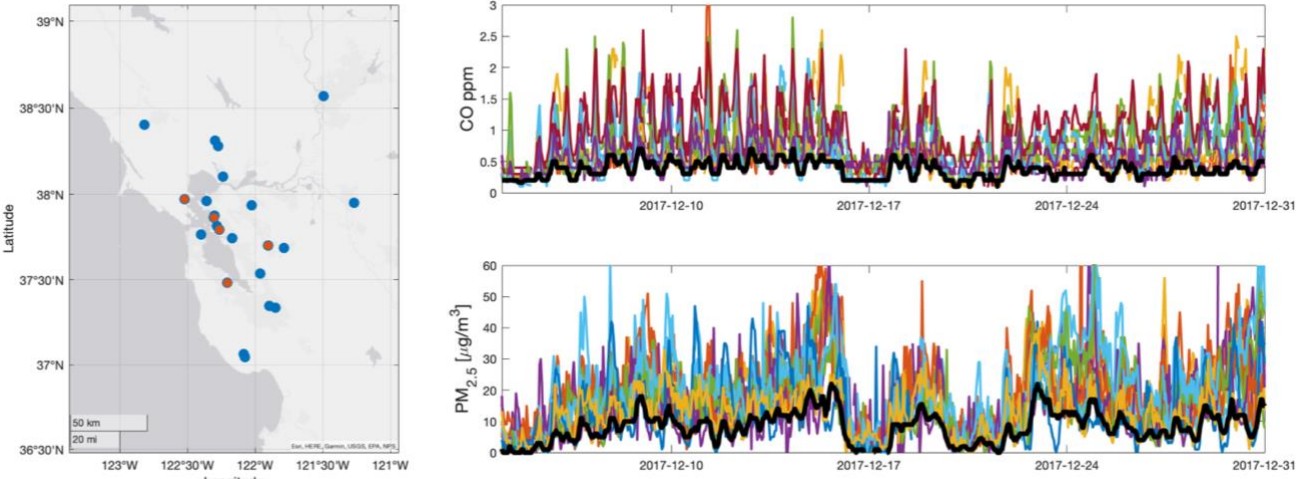

**Figure 2 (Left):** BAAQMD sites used in this study. Red dots show near-highway sites at which HDV emissions factors were determined. Blue sites were used only for determining regional signal. **Figure 2 (Right).** Aerosol and CO at each BAAQMD site (various colors). The regional background (black), is defined as the lowest 10th percentile of all signals within a rolling 4-hour window. Figure credit: Esri, HERE, Garmin, USGS, EPA, NPS.


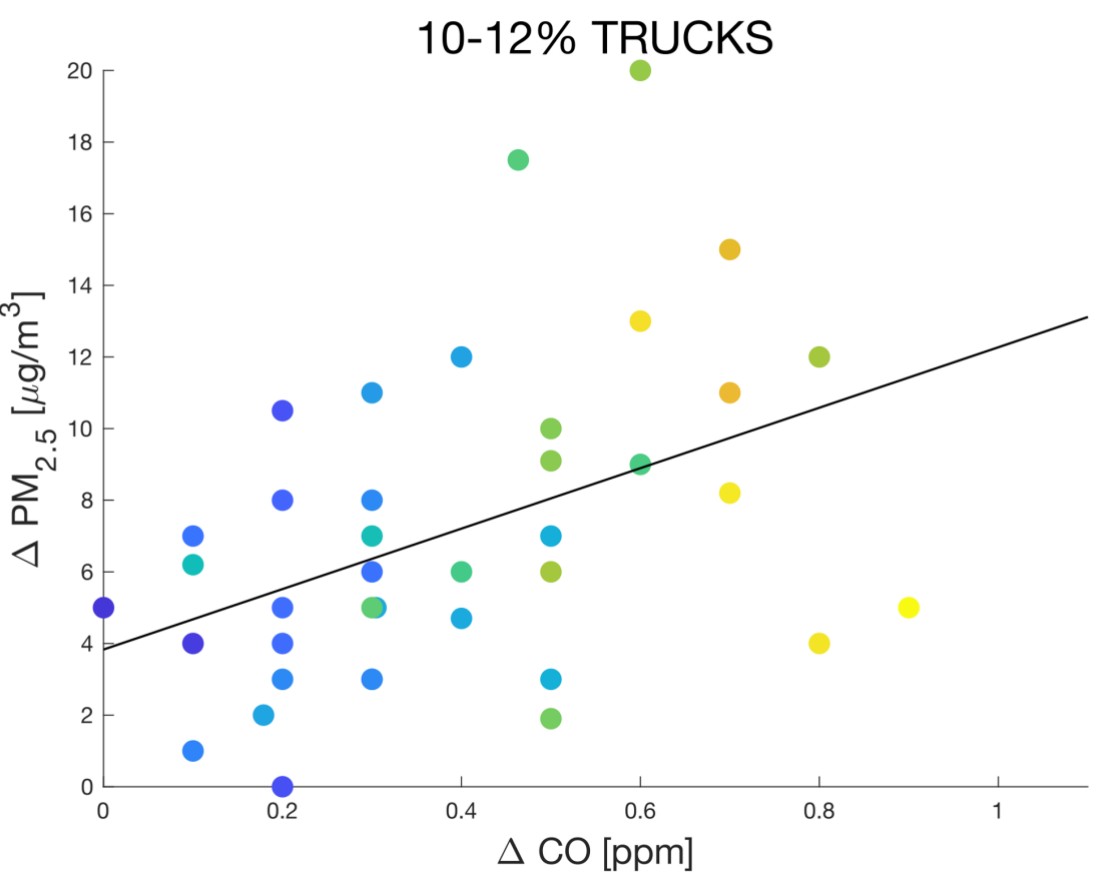


**Figure 3.** ΔPM vs. ΔCO at Pleasanton site during the 2018-2020 time period for which 10-12% of traffic flow is trucks. Data is colored by $NO_x$ concentration. These points are fit linearly to find slope.

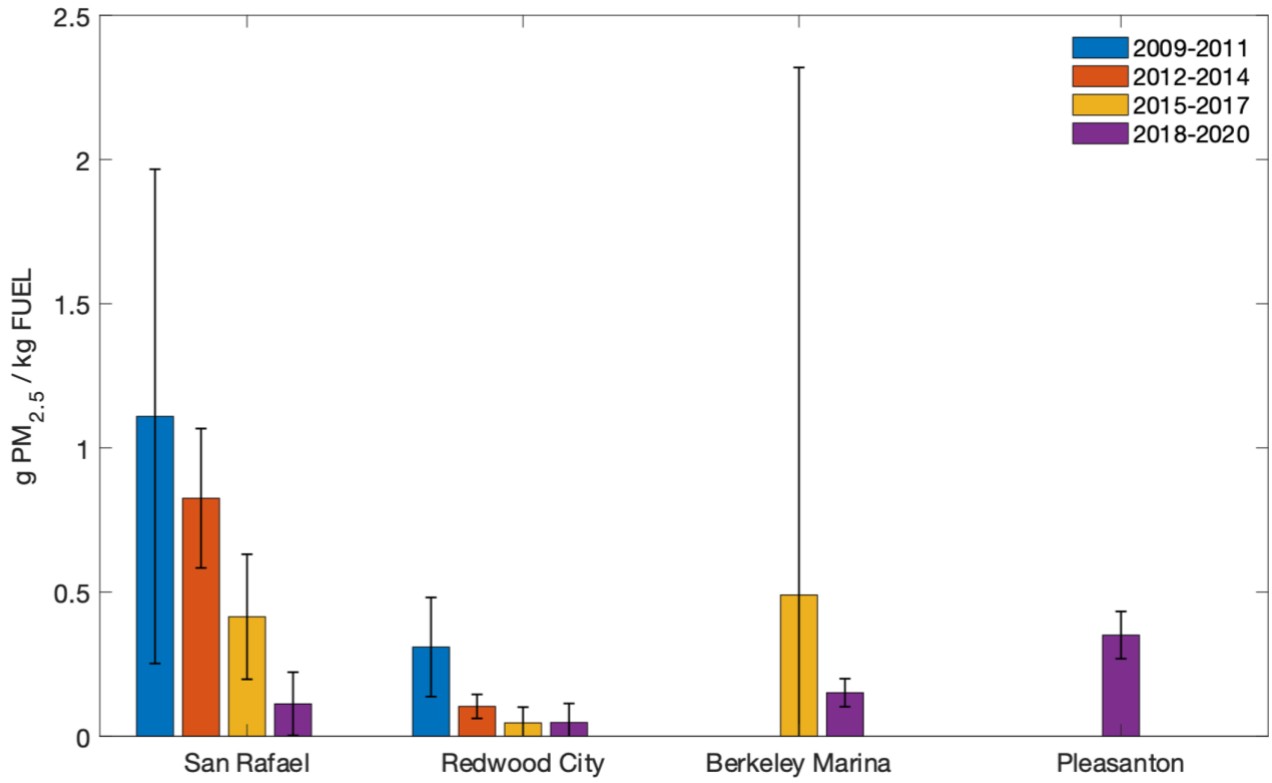

**Figure 4:** Top: Fleet emissions factors, derived from all sites, all years, binned by truck fraction. Bottom: HDV emissions factor at near highway sites during 2009-2011, 2012-2014, 2015-2017, and 2018-2020 time period.


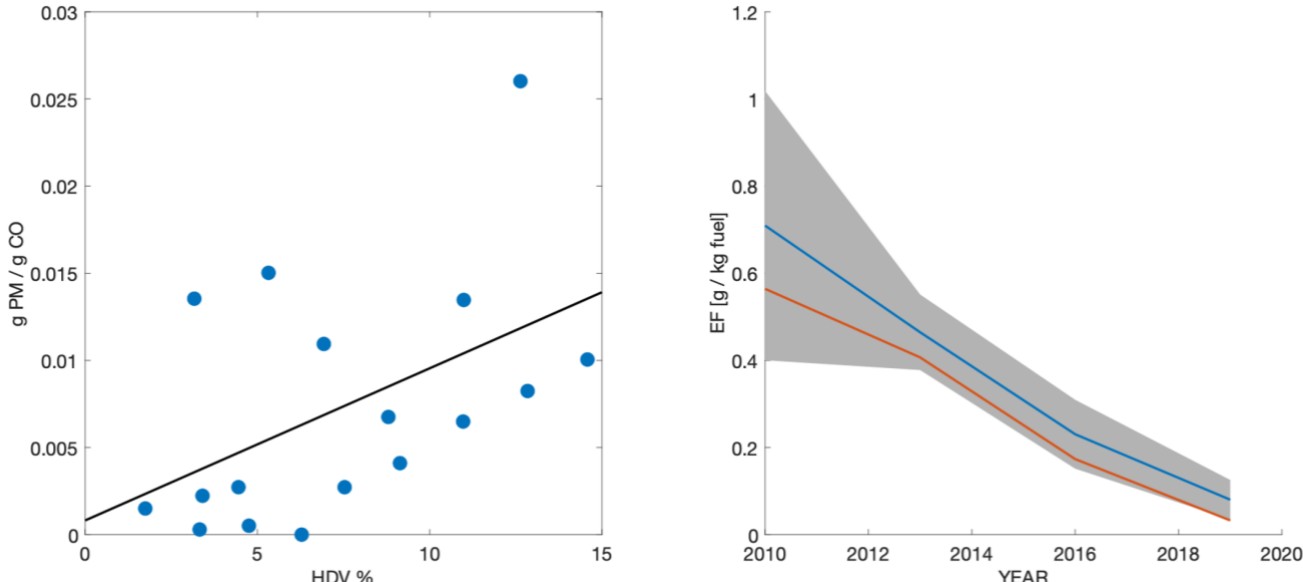

**Figure 5:** (left) Each point corresponds to the PM:CO enhancement ratio calculated via a linear fit between PM enhancement and CO enhancement at a particular near-highway site during the 2018-2020 time for each bin of HDV %. Laney College data is not included. The black line shows the linear fit corresponding to all points. (right) Trend in $EF_{PM(HDV)}$ for RWC and SR (as shown in Figure 1). The blue line indicates values calculated setting $EF_{PM(LDV)}=0$, while the orange line indicates values calculated using $EF_{PM(LDV)}= 0.002$ g PM / kg fuel. Grey patches indicate the estimated uncertainty in the error-weighted mean for the case where $EF_{PM(LDV)}=0$.

| Study | Year of Measurements | Vehicle Type | Measurement Location | $EF_{CO}$ (g/kg fuel) | $EF_{PM}$ (g/kg fuel) |
|---|---|---|---|---|---|
| *Kirchstetter (1999)* | 1997 | Light Duty | Caldecott Tunnel, Oakland CA | | $0.11 \pm 0.1$ |
| *Kirchstetter (1999)* | 1997 | Heavy Duty | Caldecott Tunnel, Oakland CA | | $2.7 \pm 0.3$ |
| *Geller (2005)* | 2004 | Light Duty | Caldecott Tunnel, Oakland CA | | $0.07 \pm 0.02$ |
| *Geller (2005)* | 2004 | Heavy Duty | Caldecott Tunnel, Oakland CA | | $1.04 \pm 0.02$ |
| *Ban-Weiss (2008)* | 2006 | Light Duty | Caldecott Tunnel, Oakland CA | | $0.07 \pm 0.2$ |
| *Ban-Weiss (2008)* | 2006 | Heavy Duty | Caldecott Tunnel, Oakland CA | | $1.4 \pm 0.6$ |
| *Park (2011)** | 2007 | Light Duty | Los Angeles, CA (Wilmington) | 47 | 0.15 |
| | 2007 | Heavy Duty | Los Angeles, CA (Wilmington) | 36 | 0.73 |
| *Dallman (2012)* | 2010 | Heavy Duty | Caldecott Tunnel, Oakland CA | 8.0 ± 1.2 | |
| *Dallman (2013)* | 2010 | Light Duty | Caldecott Tunnel, Oakland CA | 14.3 ± 0.7 | 0.038 ± 0.010 |
| *Bishop (2015)* | 2013 | Heavy Duty | Cottonwood, CA | | 0.65 ± 0.11 |
| *Bishop (2015)* | 2013 | Heavy Duty | Port of Los Angeles | | 0.031 ± 0.007 |
| *Park (2016)* | 2011 | Light Duty | West Hollywood | 15.2 ± 53.8 | 0.01 ± 0.01 |
| | 2011 | Light Duty | Boyle Heights | 36.8 ± 85.6 | 0.11 ± 0.68 |
| | 2011 | Light Duty | Los Angeles, CA (Wilmington) | 46.6 ± 117.9 | 0.04 ± 0.21 |
| *Haugen (2017)* | 2015 | Heavy Duty | Port of Los Angeles | 1.6 ± 0.4 | 0.11±0.01 |
| | 2015 | Heavy Duty | Cottonwood, CA | 3.0 ± 0.2 | 0.22 ± 0.04 |
| *Haugen (2018)* | 2017 | Heavy Duty | Port of Los Angeles | 1.7 ± 0.3 | 0.035 ± 0.01 |
| | 2017 | Heavy Duty | Cottonwood, CA | 2.8 ± 0.4 | 0.09 ± 0.005 |
| *Li (2018)* | 2014 | Light Duty | Pittsburgh, PA | | 0.035±0.008 |
| | 2014 | Heavy Duty | Pittsburgh, PA | | 0.225±0.065 |

* Note that in Park (2011), no error in emissions factors were reported.
**Table 1:** Summary of emission factors derived by previous studies.
