# Peer review of "A method for using stationary networks to observe long term trends of on-road emissions factors of primary aerosol from heavy duty vehicles"

_Atmospheric Chemistry and Physics, 2021_

## Referee Comment (RC1)

Review comments on "A method for using stationary networks to observe long term trends of on-road emissions factors of primary aerosol from heavy duty vehicles by Helen L. Fitzmaurice and Ronald C. Cohen"

This paper presents a method for determining emissions factors (EF) of primary aerosols from heavy-duty vehicles (HDV) using long-term stationary monitoring data of PM2.5, and CO. Authors combined traffic count/composition, and air pollution concentrations measured at several monitoring sites in the Bay Area to determine emission factors of PM2.5. Authors reported that estimated EFs vary substantially with time and space. The research topic is important and well suited to the scope of the journal. However, I think that the estimated emission factors using the proposed method are highly uncertain, rely on many assumptions (some of them are not very realistic, in my view). The paper is not very well written; discussions are very short; conclusions are not well substantiated by uncertainty analysis. Some of my specific comments are below.

1. The paper used ambient air pollution measurements from various sites to estimate the emission factors. They said, "We include all BAAQMD sites that are within 500 meters of one major highway and use traffic count data from the PeMS measurement site closest to each air quality site". The distances from the highway for various sites are not reported. Previous studies have used on-road or near-road ambient measurements to determine emission factors for traffic-related air pollutants. The main challenge in this process is to isolate the traffic signals from ambient measurements. Since the traffic pollution signal decay exponentially with distance from the roads, within a few meters (usually 50-100 m), traffic signals become very close to ambient/background level. If one goes away from the roadway, the decoupling of traffic and background signals becomes more and more challenging, and resulting estimates become highly uncertain. Since the roadway signals get highly diluted with downwind distance, a small error in isolating traffic versus non-traffic signals can impact emission factor estimations. This is a major limitation of this paper since they used data within 500m from the roadway.

2. The near-road signals depend on wind speed and direction and other meteorological factors. While the authors used a subset of monitoring data from morning and wind speed > 0.5 m/s, (it appears that) they did not consider wind direction. While a period with high wind speed but opposite direction, the monitoring locations will not see much highway signals. To get a good highway signal, one needs to consider wind speed and direction (and data from within a few meters of the highway).

3. Authors assumed that only HDV contributes to PM2.5. I do not fully agree with this assumption. In the current US scenario, tailpipe and non-tailpipe traffic emissions are comparable (even non-tailpipe could be higher than tailpipe) in many locations. Both HDV and LDV contribute to non-tailpipe PM emissions. Since the number of LDV in a typical highway fleet is much higher than HDV (typically 90-95% are LDV), the LDV might largely contribute to overall vehicular primary PM2.5. Also, tailpipe PM2.5 from LDV is not negligible. Therefore, when total PM2.5 is the concern, I think the assumption that only HDV contributes to PM2.5 is a wild guess.

4. Looking at Fig. 2, the estimated background PM2.5 signals (assuming 10th percentile as background) seem very uncertain. In some cases, the background PM2.5 is close to zero. As per the existing literature, the majority of PM2.5 is background. These background estimates (or decoupling highway versus roadway signal for PM2.5) are uncertain. Therefore, the resulting EFs using these data also would be highly uncertain. If they underestimate the background PM2.5 (means overestimation of traffic PM2.5), the resulting traffic EF would be higher. This could be the reason behind their estimated higher EF than other recent studies shown in Fig.1. Also, they said, "We observe an average EF of 0.11 g 145 PM / kg fuel, for 2018-2020, more than 2-3 times larger than expected for an HDV fleet compliant with current regulations". This higher estimation could be due to uncertainty in isolating traffic and background signals.

5. EF's spatial variability could also be due to the problem of isolating traffic versus non-traffic signals. If the location of a site is far away from the roadway, a small error in isolating traffic versus non-traffic signals could have a huge impact on the estimated EF. The authors tried to explain the high EF at one site based on parking lot influence. This is not very convincing. Because if one compares the number of cars on a parking lot versus a highway over a day, one expects much higher cars on a highway.

6. Equation 1 is hard to understand (it has some formatting issues). I think the details derivation of Eq. 1 is needed.

---

## Author Response (AR1)

We thank the reviewers for their thoughtful helpful comments, which we have used to guide us in making changes to the text. We address each comment below. Text from the original review is in black. Our response is in red.

**Reviewer 1:**

This paper presents a method for determining emissions factors (EF) of primary aerosols from heavy-duty vehicles (HDV) using long-term stationary monitoring data of PM2.5, and CO. Authors combined traffic count/composition, and air pollution concentrations measured at several monitoring sites in the Bay Area to determine emission factors of PM2.5. Authors reported that estimated EFs vary substantially with time and space. The research topic is important and well suited to the scope of the journal. However, I think that the estimated emission factors using the proposed method are highly uncertain, rely on many assumptions (some of them are not very realistic, in my view). The paper is not very well written; discussions are very short; conclusions are not well substantiated by uncertainty analysis. Some of my specific comments are below.

Thank you for highlighting places where we need to be more clear in justifying the assumptions we make in this paper. We use the specific comments below to adjust our methods and add explanation to the text.

1. The paper used ambient air pollution measurements from various sites to estimate the emission factors. They said, "We include all BAAQMD sites that are within 500 meters of one major highway and use traffic count data from the PeMS measurement site closest to each air quality site". The distances from the highway for various sites are not reported. Previous studies have used on-road or near-road ambient measurements to determine emission factors for traffic-related air pollutants. The main challenge in this process is to isolate the traffic signals from ambient measurements. Since the traffic pollution signal decay exponentially with distance from the roads, within a few meters (usually 50-100 m), traffic signals become very close to ambient/background level. If one goes away from the roadway, the decoupling of traffic and background signals becomes more and more challenging, and resulting estimates become highly uncertain. Since the roadway signals get highly diluted with downwind distance, a small error in isolating traffic versus non-traffic signals can impact emission factor estimations. This is a major limitation of this paper since they used data within 500m from the roadway.

By focusing our analysis on the early morning, when the boundary layer height is low, we observe highway signals at distances of up to ~500 m. While it is true that other studies have demonstrated that substantial signal decay occurs within 50-100 m, these studies report data that was collected later in the day, when both the boundary layer height is higher and vertical mixing more vigorous. Previous studies support this conclusion that the length scales vary with

time-of-day. For example, Choi et al. (2014) show that enhancement decreases on length scales ~500-1000 meters from a highway and they report persistent enhancements up to 2 km away.

2. The near-road signals depend on wind speed and direction and other meteorological factors. While the authors used a subset of monitoring data from morning and wind speed > 0.5 m/s, (it appears that) they did not consider wind direction. While a period with high wind speed but opposite direction, the monitoring locations will not see much highway signals. To get a good highway signal, one needs to consider wind speed and direction (and data from within a few meters of the highway).

Thank you for this suggestion. We have implemented wind filtering and it has not substantially changed our results or analysis. (See description of wind filtering L140 and table below in response to Reviewer 2's suggestion around fitting method.)

3. Authors assumed that only HDV contributes to PM2.5. I do not fully agree with this assumption. In the current US scenario, tailpipe and non-tailpipe traffic emissions are comparable (even non-tailpipe could be higher than tailpipe) in many locations. Both HDV and LDV contribute to non-tailpipe PM emissions. Since the number of LDV in a typical highway fleet is much higher than HDV (typically 90-95% are LDV), the LDV might largely contribute to overall vehicular primary PM2.5. Also, tailpipe PM2.5 from LDV is not negligible. Therefore, when total PM2.5 is the concern, I think the assumption that only HDV contributes to PM2.5 is a wild guess.

Thank you for emphasizing this important point. We add a paragraph (L188-L205) discussing this assumption and the evidence supporting it. To clarify the issue, we have brought Fig. 5 which addressed this issue in the supplement forward into the main text. There is a significant (order of magnitude) disagreement about LDV emission factors in the literature, especially for non-tailpipe emissions (Fussell, et al., 2022). However, our data (Figure 5 left) shows that PM2.5:CO enhancement ratios increase in proportion to HDV %, with a small intercept (Fig. 5, left). This correlation is our evidence for HDVs dominating emissions. We also re-compute HDV emissions factors using a fixed LDV emission factor (Fig. 5, right).

4. Looking at Fig. 2, the estimated background PM2.5 signals (assuming 10th percentile as background) seem very uncertain. In some cases, the background PM2.5 is close to zero. As per the existing literature, the majority of PM2.5 is background. These background estimates (or decoupling highway versus roadway signal for PM2.5) are uncertain. Therefore, the resulting EFs using these data also would be highly uncertain.

If they underestimate the background PM2.5 (means overestimation of traffic PM2.5), the resulting traffic EF would be higher. This could be the reason behind their estimated higher EF than other recent studies shown in Fig.1. Also, they said, "We observe an average EF of 0.11 g 145 PM / kg fuel, for 2018-2020, more than 2-3 times larger than expected for an HDV fleet compliant with current regulations". This higher estimation could be due to uncertainty in isolating traffic and background signals.

An error in the background of the sort the reviewer described would not scale with % HDV. (See Fig. 5.) Such an error would be a constant offset affecting the intercept of our analysis and be attributed to LDVs. We add discussion surrounding the impact of LDV emissions to Sect. 3 (L188-L205).

5. EF's spatial variability could also be due to the problem of isolating traffic versus non-traffic signals. If the location of a site is far away from the roadway, a small error in isolating traffic versus non-traffic signals could have a huge impact on the estimated EF. The authors tried to explain the high EF at one site based on parking lot influence. This is not very convincing. Because if one compares the number of cars on a parking lot versus a highway over a day, one expects much higher cars on a highway.

It is true that high EFs may be related to isolating highway traffic from non-traffic or non-highway traffic signals, and we add a qualification about our results to this affect (L206-L209; L216-218). However, even at sites with high calculated EFs (such as Pleasanton in 2018-2020), we see increasing NOx with increasing PM$_{2.5}$ and CO enhancement.

With regards to the Laney College site, vehicles in a parking lot drive much more slowly than on the highway. As discussed in Sect. S8, emission models predict a 40 times higher PM$_{2.5}$:CO ratio for LDV driving at 5mph compared to 50 mph. We model the impact of ~650 cars per hour driving through a parking lot and show that the added PM$_{2.5}$ from these LDV explain a substantial portion of the difference between what is observed and what would be expected using EMFAC2017 emission factors, given observed truck volumes (Fig. S8.)

6. Equation 1 is hard to understand (it has some formatting issues). I think the details derivation of Eq. 1 is needed.

We now detail the derivation in Sect. S3.

$$EF_{PM,HDV} = \frac{g\ PM_{HDV}}{kg\ fuel_{HDV}}.$$

We multiply this expression by $\frac{g\ CO_{fleet}}{g\ CO_{fleet}}$ and $\frac{kg\ fuel_{fleet}}{kg\ fuel_{fleet}}$, getting:

$$EF_{PM,HDV} = \frac{g\ PM_{HDV}}{kg\ fuel_{HDV}} \frac{g\ CO_{fleet}}{g\ CO_{fleet}} \frac{kg\ fuel_{fleet}}{kg\ fuel_{fleet}}.$$

Rearranging, we find:

$$EF_{PM,HDV} = \frac{g\ PM_{HDV}}{g\ CO_{fleet}} \frac{g\ CO_{fleet}}{kg\ fuel_{fleet}} \frac{kg\ fuel_{fleet}}{kg\ fuel_{HDV}}, \text{ so}$$

$$EF_{PM,HDV} = \frac{g\ PM_{HDV}}{g\ CO_{fleet}} \frac{g\ CO_{fleet}}{kg\ fuel_{HDV}}.$$

Because we measure concentrations of PM$_{2.5}$ (µg m$^{-3}$) and CO (ppm) rather than g PM emitted and g CO emitted, we convert using the ideal gas law.

$$EF_{PM,HDV} = \gamma \, \frac{PM_{HDV}}{CO_{fleet}} \, \frac{g \, CO_{fleet}}{kg \, fuel_{HDV}}.$$

We calculate $\gamma$ is using the ideal gas law, assuming STP.

We have added a section to our supplement detailing this derivation, as we do not think that it fits well within the narration of the main text.

**Reviewer 2:**

In this manuscript, the authors calculated the on-road emission factors of heavy-duty vehicles (HDV) in the San Francisco Bay area using BAAQMD's ambient monitoring data. The results show that the HDV emission factors decreased by a factor of 7 in the past decades, which is in line with other near-road and tunnel observations in the US. And the authors also found that the HDV emission factors have large spatial variations. The monitoring data from BAAQMD's monitoring network was also used to estimate people's exposure to primary PM2.5 from HDV emissions in this study. Overall, I think the method developed by the authors is potentially useful and can be applied to other EPA near-road stations to estimate HDV emission factors around the US. However, the emission factors estimated by this method are highly uncertain, and the authors haven't fully characterized the uncertainty associated with this method.

Thank you for these comments. We have used your suggestions below to further characterize the uncertainties associated with our method.

1. Since the time resolution of the monitoring data is very low (1-h), it is challenging to separate the HDV emissions from the background, and the choice of background concentrations can significantly affect the results. In this study, the authors used the 10th percentile of all measurements collected within a 5-hour window across the entire San Francisco Bay area as the background, which seems arbitrary.

We include a sensitivity test to the time-window chosen in section S4.

The authors need to run more sensitivity tests about the background concentration. How different would the emission factors be if another percentile was chosen as background?

We added tests of the sensitivity of the derived HDV emission factor to the inferred background concentration, by using the 5$^{th}$, 10$^{th}$, 15$^{th}$, 20$^{th}$, and 25$^{th}$ percentile to calculate background concentration. We find that while changing the percentile results in differences to the estimated HDV emission facto that are small in comparison to year over year differences. We have added a section S4 to the supplement, discussing this analysis.

For each near-road station, if you only use concentrations measured at the closest station or the lowest concentration measured at stations within a closer distance (like 10 km), how different would the calculated HDV emission factor be?

Most stations are greater than 10 km from one another, meaning this method would not be practical to implement for the BAAQMD network. This would be interesting to explore further in the case of denser networks such as Purple air or BEACO$_2$N (Shusterman et al., 2016; Shusterman et al., 2018; Kim et al., 2018; Kim et al., 2022).

2. For the background-corrected PM2.5-to-CO ratio shown in Figure 3, the authors should do the fitting using the original data instead of binning the CO concentration. By binning data, a tiny portion of data in the high delta CO range (>0.8 ppm) is dragging the overall fitting.

As suggested, we now use all the original data not filtered by wind or fire criteria in the fits. We initially used medians of bins to eliminate the impact of noise we thought to be from non-highway sources. However, by implementing a wind filter as suggested above, this noise was reduced, so when combined with the addition of a wind filter, fitting all data instead of binned data ahs little impact on the derived emission factors. We include a comparison table here. Un-highlighted values are from our original method, using median point values only in fitting. Highlighted numbers are generated through slopes found fitting all data (and wind filtering), as now shown in Fig. 4. (All values are HDV EF estimates in g PM$_{2.5}$ / kg fuel.)

With the exception of Redwood City in 2009-2011 and San Rafael 2018-2020, these numbers match to within current error estimations (Fig. 4). In both of these cases, original values were higher, possibly indicating a contribution from nearby non-highway sources. Using all data points instead of bins allows us to estimate an emission factor for Berkeley Marina in 2015-2017 as well, although the estimated uncertainty is large relative to the estimated emission factor. Because this site came online during the 2015-2017 period, by using the binning method, we did not have enough points to fit a line.

| TIME PERIOD | SAN RAFAEL | REDWOOD CITY | BERKELEY MARINA | PLEASANTON |
|---|---|---|---|---|
| 2009-2011 | 0.98 | .48 | N/A | N/A |
| | 1.10 | 0.31 | N/A | N/A |
| 2012-2014 | 0.94 | 0.08 | N/A | N/A |
| | 0.86 | 0.10 | N/A | N/A |
| 2015-2017 | 0.32 | 0.05 | N/A | N/A |
| | 0.42 | 0.05 | 0.50 | N/A |
| 2018-2020 | 0.21 | 0.02 | 0.17 | 0.38 |
| | 0.11 | 0.05 | 0.15 | 0.36 |

The authors should also estimate the uncertainty associated with this fitting and propagate it to the overall uncertainty range.

We now show the uncertainty in the fitting in S5. We propagate this uncertainty, as described in Sect. S6, and use this uncertainty propagation to add error bars to Figure 4.

3. The authors need to thoroughly discuss uncertainties associated with all terms in Equation 1 and 2 and propagate them to the results.

We add a discussion of the uncertainties associated within each term, as well as the propagation of these uncertainties to the supplement. (See Section S6.) We use the described uncertainty propagation to characterize uncertainty in the emission factors we show in Figure 4.

4. The emission factors in Figure 4 should have uncertainty bars. Because the method has large uncertainties from the choice of background concentrations, the spatial variation estimated using this method may not be real. How were the traffic speed and slope of the road at those near-road stations? The spatial variation may also be caused by traffic speed and road slope.

We add uncertainty bars to the emission factors in Figure 4 as discussed in response to previous comment. We agree that on-road factors such as traffic speed and road slope may have a substantial impact on emission factors. None of the lengths of roadway in Figure 4 are subject to a substantial grade. We incorporate day-to-day variance in traffic speed into our new uncertainty calculation.

5. Did the authors try analyzing the monitoring data around noontime? The HDV traffic is usually the highest around noontime.

We do not try analyzing the data at noontime, because by that time the boundary layer height is substantially larger than during the AM rush hour, meaning that emissions are likely to be substantially more dilute before reaching BAAQMD monitoring sites than in the AM. While HDV emissions may be slightly higher at noontime than during AM rush hour, they are not substantially so (< 25% higher for all sites examined).

6. The wind speed and wind direction data are also measured at BAAQMD's monitoring stations. Why did the authors use wind data from the reanalysis product instead of the measurements at monitoring stations?

We use the ECMWF reanalysis product instead of the measurements at BAAQMD monitoring stations, because the meteorological measurements at the BAAQMD monitoring stations are unreasonably difficult to access. While BAAQMD posts meteorological data to its website, to the best of our knowledge, there is no API for BAAQMD meteorological measurements that we could find. For example, while BAAQMD air quality measurements can be downloaded using the EPA's API service, wind speed and wind direction are not available via this service.

7. The authors should be more careful about using parameters derived from the EMFAC model to calculate on-road HDV emissions. The emission factors estimated by the authors are under the situation when HDVs are driving on-road at a certain speed with a particular road slope. However, the emission factors modeled by EMFAC consider the entire driving cycle, different seasons, different types of fuels, and all driving conditions. The authors should provide more details about how they ran the EMFAC model.

This is a good point, as both fuel efficiency and emission factors from other pollutants can vary considerably as a function of specific driving conditions. We have created a new methods section (2.4) in which we detail how we run the EMFAC model to estimate CO emission factors as well as emission rates (g $CO_2$ / vkm). The methods we use follow those in Fitzmaurice et al., 2022. We also add the impact of speed variance on emission factors to our estimation of uncertainty in HDV PM emission factors.

**References:**

Choi, W., Winer, A.M. and Paulson, S.E., 2014. Factors controlling pollutant plume length downwind of major roadways in nocturnal surface inversions. *Atmospheric Chemistry and Physics*, *14*(13), pp.6925-6940., https://doi.org/10.5194/acp-14-6925-2014

Fitzmaurice, H.L., Turner, A.J., Kim, J., Chan, K., Delaria, E.R., Newman, C., Wooldridge, P. and Cohen, R.C., 2022a. Assessing vehicle fuel efficiency using a dense network of CO 2 observations. *Atmospheric Chemistry and Physics*, *22*(6), pp.3891-3900.

Fussell, J.C., Franklin, M., Green, D.C., Gustafsson, M., Harrison, R.M., Hicks, W., Kelly, F.J., Kishta, F., Miller, M.R., Mudway, I.S. and Oroumiyeh, F., 2022. A Review of Road Traffic-Derived Non-Exhaust Particles: Emissions, Physicochemical Characteristics, Health Risks, and Mitigation Measures. *Environmental Science & Technology*.

Kim, J., Shusterman, A. A., Lieschke, K. J., Newman, C., & Cohen, R. C. The Berkeley Atmospheric CO2 Observation Network: Field calibration and evaluation of low-cost air quality sensors. Atmospheric Measurement Techniques, 11(4), 1937–1946. https://doi.org/10.5194/amt 11-1937-2018, 2018.

Kim, J., Turner, A.J., Fitzmaurice, H.L., Delaria, E.R., Newman, C., Wooldridge, P.J. and Cohen, R.C., 2022. Observing Annual Trends in Vehicular CO2 Emissions. *Environmental Science & Technology*. https://doi.org/10.1021/acs.est.1c06828

Shusterman, A. A., Teige, V. E., Turner, A. J., Newman, C., Kim, J., & Cohen, R. C., The Berkeley Atmospheric CO2 Observation Network: Initial evaluation. Atmospheric Chemistry and Physics, 16(21), 13449–13463. https://doi.org/10.5194/acp-16-13449-2016, 2016.

Shusterman, A. A., Kim, J., Lieschke, K. J., Newman, C., Wooldridge, P. J., & Cohen, R. C. (2018). Observing local CO2 sources using low-cost, near-surface urban monitors. *Atmos. Chem. Phys*, *18*, 13773-13785.

---

## Author Response (AR2)

**Response to Referee 1:**

Referee 1 raises two important concerns about our methods.
1. Contributions of LDV to PM Emissions
2. Goodness of fit for PM:CO enhancement ratios

**Concern 1: Contributions of LDV to PM Emissions**
Reviewer 1 is concerned about our assumption that HDVs are responsible for the bulk of $PM_{2.5}$ emissions, citing Habre et. al, 2020, which found LDVs responsible for a large percentage of $PM_{2.5}$ emissions in the LA region. Although this is an important concern, the methodology use (and consequently findings) and context examined by Habre et. al, 2020 differ enough from the methodology and context of this work, that we question the relevance of this paper's findings to our study.

**Methodological Differences:** Unlike the work we present, Habre et. al, 2020 does not use hourly filter measurements measuring total $PM_{2.5}$ mass, but instead uses filter measurements collected over full months (via two two-week samples) for their analysis. Habre et al., 2020 then use speciation data from filter samples in combination with traffic factors, to run the EPA Positive Matrix Factorization (EPA PMF v5.0) model. Using this model, Habre et al., 2020 estimate fraction of total aerosol burden attributable to various roadway sources for three size fractions, quasi-ultrafine (0.-.2 μm), fine (.2 - 2.5 μm), and coarse (2.5 -10 μm), and are able to explain 63%, 86%, and 88% of the variability in the quasi-ultrafine, fine, and coarse aerosol mass. Through this modeling Habre et al., 2020 estimate that LDV contribute 72% of tailpipe emissions in the fine size fraction by mass, and that abrasive vehicle emissions (not differentiated as HDV or LDV) make up 35% of the fine size fraction of traffic emissions by mass. However, these factors together make up only 32% of the explained fine aerosol mass and Habre et al., 2020 do not provide error estimates on these percentages. Furthermore, although Habre et al., 2020 quantifies the relative burden of LDV v. HDV tailpipe emissions, they do not attempt to calculate emission factors for either, making it difficult for us to compare their results to our own.

**Contextual Differences:** As opposed to our work, which focuses on spatially and temporally on highway emissions, Habre et al., 2020 analyzes aerosol measurements over wide areas, meaning that their analysis is meant to include emissions from surface streets, where vehicles tend to drive more slowly but brake more frequently. There are a number of reasons that we expect the relative contribution of LDV and HDV to aerosol mass to be different on highways v. surface streets. First, the percentage of vehicles that are HDV on surface streets is likely lower than what would be expected on highways. For example, in the SF Bay Area, EMFAC2017 estimates HDV fraction of all traffic driving at 30 mph to be ~0.7%, compared to ~4.4% for vehicles driving 60 mph. Furthermore, Habre et al., 2020 excludes data from the HDV-rich area of Long Beach in drawing conclusions related to the relative contributions to aerosol burden of HDV as compared to LDV. Because Habre et al., 2020 does not report the percentage of HDV in the area studied or attempt to quantify HDV or LDV emission factors, making a direct comparison difficult.

Beyond the contribution of HDV v. LDV to tailpipe aerosol emissions, Reviewer 1 is also concerned that a large fraction of highway PM emissions may be from abrasive vehicle emissions (AVE), including tire wear and brake wear. We do not dispute this idea, especially in recent years as HDV tailpipe controls have been put in place, substantially reducing emissions from HDV tailpipes, and we have clarified the text of our paper (L181-182) to make that more clear. Reviewer 1 is concerned that a large portion of AVE may be from LDV. However, on a per vehicle basis, HDV are expected to contribute significantly more to AVE. Estimates of the ratio of LDV and HDV AVE emission factors from previous studies vary widely and often consider $PM_{10}$ or total mass loss (rather than $PM_{2.5}$) due to tire or brake wear. For example, MOVES3 Break and Tirewear Emissions (EPA, 2020) finds a wide range of total tire wear mass ratios per mile (HDV:LDV), ranging from 2.3 (Bauman et al., 1997) to 26 (Senco,

1999), with most estimates lying (Garben et al, 1997;  Gebbe et al., 1997; EMPA, 2000)  in the low to mid teens. Generally, tire wear is expected to increase with tire number and weight. HDV have ~4x more tires and ~15x the weight of LDV. The MOVES3 Brake and Tirewear Emissions (EPA, 2020) finds similarly large range of emission factor ratio estimates for brake wear (EPA, 2020). Only one HDV:LDV emission factor ratio was reported for $PM_{2.5}$. Abu-Allaban et al., 2003 reported a value of 3. However, for $PM_{10}$, the HDV:LDV emission factor ratio reported by previous literature ranges from 0.7 (Carbotech, 1999) to 24 (Rauterburg-Wulff, 1999) with most estimates (Luhana et al., 2004; Abu-Allaban et al., 2003; Westurland, 2001) in the range of 6-8.

Tailpipe emissions are similarly expected to be much higher on a per vehicle basis for HDV compared to LDV. In the studies we cite in Table 1 of the text, Geller et al., 2005 reported  an HDV: LDV $PM_{2.5}$ emission rate ratio of 14, Ban-Weiss et al., 2008 reported a ratio of 20, and Li et al., 2014 reported a ratio of 7.5. Park et al., 2011 reported a lower ratio of 4.8, although we note that this was on surface streets rather than a highway. Similar to the studies we cite, at 60 mph, the EMFAC2017 model yields a HDV: LDV $PM_{2.5}$ emission rate ratio of 23 in 2009 and 8 in 2020.

Using literature-reported emission factor ratios, we put our assumption that HDV are responsible for the bulk of $PM_{2.5}$ emissions into context by considering the fraction of PM expected to be from HDV as a function of emission factor ratios. Ultimately, the validity of assumption that HDV contribute the bulk of aerosol is dependent on the ratio between HDV and LDV emission factors. In Fig. R1, we show the percentage of vehicle PM we expect to be from HDV on a highway as a function of HDV% for three scenarios: EFratio = 1, EFratio = 8, EFratio = 10, EFratio = 100, where the EFratio is defined as $EF_{PM(HDV)}$ / $EF_{PM(LDV)}$. Note that even when EFratio = 1, because emissions factors are in g PM / kg fuel, the percent of PM from HDV rises faster than HDV % because HDV burn 3-4x fuel per distance traveled. In 2009, we expect real emissions to fall between the EFratio = 10 and EFratio = 100, while in 2020, we expect real emissions to be closer to the EFratio = 8 line.

Finally, we feel that the evidence we present in Figure 5: a linear correlation between PM:CO enhancement ratio and HDV fraction with a very small intercept, and an estimated change in emission factors resulting from the introduction of a constant LDV emission factor, show that our assumptions are reasonable.

**Concern 2: Goodness of fit for PM:CO enhancement ratios**

Referee 1 is concerned about the statistics of the fitting we use in finding the ratio of PM enhancement to CO enhancement. Particularly, they are concerned about low $R^2$ values for our fits. We agree, as reviewer 1 suggests, that on an hourly basis, there is substantial uncertainty in calculating enhancements of PM and CO. We also concede that using a 68% CI to estimate uncertainty in the slopes we find to estimate error in emission factors may provide an overly optimistic picture of certainty in our estimates of emission factors. To address this, we have replaced the 68% CI with a 95% CI in slope to estimate the uncertainty in the fits used to estimate emission factors and carry this larger (95% CI) uncertainty for each slope through the process used to estimate emission factors.

However, we disagree with Referee 1's view that the noise in our data makes the fits we find essentially meaningless and totally uncertain. The information contained in this noisy data can be seen more clearly in considering median PM enhancement for bins of CO enhancement. When we fit binned medians, instead of all points, $R^2$ values increase substantially (Table R1), highlighting the information contained in this noisy data. Furthermore, when these binned medians are used to estimate emission factors, all emission factors agree to within error of emission factors estimated using all points (Table R2). Note that these values are slightly different than the table reported in the previous round of responses, as the median fit values from the previous round used all data before wind filtering and we now use data that has had a wind filter applied for

both fit types. In the manuscript, we retain the "all points" fit as it was requested by reviewer 2, in the previous iteration of reviews.

We also note that $R^2$ values are worse when considering lower slopes. However, we find several cases in which $R^2$ values are relatively low, even for the binned median fits, but points fit the line well. See example in Figure R2. For fits to low slopes, and consequently small estimated emission factors, our estimated uncertainty in our factors is often large relative to the size of the estimated emission factor, and we believe the confidence interval of the slopes and consequent uncertainty estimates in emission factors to be reasonable.

**Response to Referee 2:**

We now describe the shaded region in Figure 5. In both Figure 1 and Figure 5, shaded region represents an error estimate for the blue line showing emissions factors from this study calculated by using error in quadriture.

**References*:**

* Carbotech, 1999 and Westerlund, 2001 were cited in a table EPA, 2020's *Brake and Tire Wear Emissions from Onroad Emissions for MOVES3,* but we were unable to find the original reference.

Abu-Allaban, M., Gillies, J.A.,Gertler,A.W., Clayton ,R., Proffitt,D. (2002). Tailpipe, re-suspended road dust, and brake wear emission factors from on-road vehicles. AtmosphericEnvironment, 37(1),5283-5293.

EPA, 2020. *Brake and Tire Wear Emissions from Onroad Emissions for MOVES3*. Retrieved from https://nepis.epa.gov/Exe/ZyPDF.cgi?Dockey=P1010M43.pdf, October 23, 2022

Habre, R., Girguis, M., Urman, R., Fruin, S., Lurmann, F., Shafer, M., Gorski, P., Franklin, M., McConnell, R., Avol, E. and Gilliland, F., 2021. Contribution of tailpipe and non-tailpipe traffic sources to quasi-ultrafine, fine and coarse particulate matter in southern California. *Journal of the Air & Waste Management Association*, *71*(2), pp.209-230.

EMPA (2000). Anteil des Strassenverkehrs an den PM10 und PM2.5 Imissionen. NFP41, Verkehr und Umwelt, Dubendorf, Switzerland.

Garben et al. (1997). Emissionkataster Kraftfahrzeugverkehr Berlin 1993, IVU GmbH Berlin, Gutachten im Auftrag der Senatsverwaltung fur Stadtenwicklung, Umweltschutz undTechnologie, Berlin, unveroeffentlich.

Gebbe et al. (1997). Quantifizierung des Reifenabriebs von Kraftfahrzeugen in Berlin, ISS-Fahrzeugtechnik, TU Berlin, i.A. der Senatsverwaltung fur Stadtenwicklung, Umweltschutz und Technologie, Berlin.

SENCO (Sustainable Environment Consultants Ltd.) (2000). Collation of information on particulate pollution from tyres, brakes, and road surfaces. 23 March, 1999, Colchester, Essex, UK

Rauterberg-Wulff A. (1999). Determination of emission factors for tyre wear particles up to 10μm by tunnel Measurements. Proceedings of 8th International Symposium 'Transport and Air Pollution'.

Luhana, L., Sokhi, R., Warner, L., Mao, H , Boulter, P., McCrae, I.S., Wright, J. Osborn, D, (2004). Non-exhaust particulate measurements:results, Deliverable 8 of the European Commission DG TrEn, 5th Framework PARTICULATES project , Contract No. 2000 - RD.11091, Version 2.0.

[Figure]

**Figure R1:** Expected fraction of PM from HDV (compared to total PM), for four different EF ratios, assuming fuel efficiencies 25 mpg for LDV and 7 mpg for HDV.

[Figure]

**Figure R2:** PM and CO enhancements for Redwood City in 2015-2027 for 4-6% trucks. Yellow points represent hourly data. Black dots represent binned medians. Fit values on the left represent fit to binned medians. Fit values on the right represent fit to all points. Numbers in parentheses represent 95th confidence intervals for all fit parameters.

| | San Rafael | | Redwood City | | Berkeley Marina | | Pleasanton | |
|---|---|---|---|---|---|---|---|---|
| **2009 - 2011** | 0.02 | 0.41 | 0.02 0.03 | 0.02 0.34 | | | | |
| **2012 - 2014** | 0.10 0.03 | 0.49 0.77 | 0.06 0.05 0.06 | 0.29 0.55 0.19 | | | | |
| **2015 - 2017** | 0.01 0.003 | 0.09 0.30 | 0.02 0.002 | 0.22 0.10 | | | | |
| **2018 - 2020** | 0.0002 0.03 | 0.51 0.27 | 0.024 0.002 | 0.68 0.05 | 0.06 | 0.82 | 0.25 0.31 0.18 | 0.40 0.86 0.57 |

**Table R1:** Here we report $R^2$ values for fits between PM enhancement and CO enhancement for each location and time period. Values on the left reflect $R^2$ for fits using all points, while those on right (yellow) reflect $R^2$ for fits using binned medians.

|  | San Rafael | Redwood City | Berkeley Marina | Pleasanton |
|---|---|---|---|---|
| **2009 - 2011** | 1.10 +/- 0.86
**2.02 +/- 1.63** | 0.31 +/- 0.17
**0.41 +/- 0.28** |  |  |
| **2012 - 2014** | 0.83 +/- 0.24
**0.72 +/- 0.29** | 0.10 +/ 0.04
**0.15 +/- 0.06** |  |  |
| **2015 - 2017** | 0.41 +/- 0.21
**0.15 +/- 0.19** | 0.05 +/- 0.05
**0.08 +/- 0.08** | 0.49 +/- 1.80
—--- |  |
| **2018 - 2020** | 0.11 +/- 0.11
**0.15 +/- 0.12** | 0.05 +/- 0.06
**0.04 +/- 0.05** | 0.15 +/- 0.05
**0.16 +/- 0.06** | 0.35 +/- 0.08
**0.24 +/- 0.11** |

**Table R2:** In this table, we calculate emission factor values calculated using either all points (top, not bold) or binned medians (**bottom, bold**). All values are in units g $PM_{2.5}$ / kg fuel.